# Transcription Factors in Cancer: When Alternative Splicing Determines Opposite Cell Fates

**DOI:** 10.3390/cells9030760

**Published:** 2020-03-20

**Authors:** Silvia Belluti, Giovanna Rigillo, Carol Imbriano

**Affiliations:** Department of Life Sciences, University of Modena and Reggio Emilia, via Campi 213/D, 41125 Modena, Italy; silvia.belluti@unimore.it (S.B.); giovanna.rigillo@unimore.it (G.R.)

**Keywords:** transcription factors, alternative splicing, cancer, cell proliferation, cell differentiation, cell death

## Abstract

Alternative splicing (AS) is a finely regulated mechanism for transcriptome and proteome diversification in eukaryotic cells. Correct balance between AS isoforms takes part in molecular mechanisms that properly define spatiotemporal and tissue specific transcriptional programs in physiological conditions. However, several diseases are associated to or even caused by AS alterations. In particular, multiple AS changes occur in cancer cells and sustain the oncogenic transcriptional program. Transcription factors (TFs) represent a key class of proteins that control gene expression by direct binding to DNA regulatory elements. AS events can generate cancer-associated TF isoforms with altered activity, leading to sustained proliferative signaling, differentiation block and apoptosis resistance, all well-known hallmarks of cancer. In this review, we focus on how AS can produce TFs isoforms with opposite transcriptional activities or antagonistic functions that severely impact on cancer biology. This summary points the attention to the relevance of the analysis of TFs splice variants in cancer, which can allow patients stratification despite the presence of interindividual genetic heterogeneity. Recurrent TFs variants that give advantage to specific cancer types not only open the opportunity to use AS transcripts as clinical biomarkers but also guide the development of new anti-cancer strategies in personalized medicine.

## 1. Introduction

### 1.1. Alternative Splicing

Alternative Splicing (AS) occurs when two or more splice sites compete and generate different transcripts from the same gene. In humans, about 95% of multiexonic genes are alternatively spliced [1]: the resulting mRNA isoforms can be differently regulated in terms of localization, stability, translational efficiency and, if translated into proteins, can give rise to isoforms with different structure and function. 

Aberrant or misregulated RNA splicing events contribute to cancer cell phenotypes by directly or indirectly regulating cell genome, epigenome, transcriptome and proteome [2,3,4]. Several studies described alterations in splicing events related to cancer, although it is not easy to identify whether proteins produced by altered AS are causative of cell transformation or are the consequence of perturbed mRNA processing [5,6]. AS occurs through different mechanisms that majorly consist of inclusion of exons, mutually exclusive exons, intron retention and alternative 5′ or 3’ splice sites. Protein isoforms produced by AS of the same gene contribute to proteome complexity and may have diverse or even opposite biological functions, with severe consequences on cellular processes and phenotypes. AS can occur also in genes encoding for transcription factors (TFs), therefore expanding their regulatory capacities for transcriptional gene control. 

### 1.2. Transcription Factors

Transcription factors (TFs) are DNA-binding proteins that control chromatin structure and gene transcription, mostly by influencing the formation or the activity of the transcriptional machinery. The majority of TFs have a sequence-specific DNA-binding domain (DBD), through which they directly bind DNA elements within transcriptional regulatory regions, such as promoters and enhancers. Besides, an activation/repression domain allows the interaction with other cofactors. Based on the type of DBD, TFs are classified in superclasses that are helix-turn-helix, zinc-coordinating, zipper-type, other α-helix, β-sheet, β-hairpin/ribbon and enzymes, this last one being an exception to the classification criterion and including proteins with enzymatic activity when bound to DNA [7]. Chromatin immunoprecipitation sequencing (Chip-seq) highlighted that the presence of TF binding motifs does not necessarily predict TF occupancy and activity in vivo [8,9]. Indeed, multiple aspects can modulate TF genomic binding, such as TF tissue-specific expression levels, dose-dependent effects, combinatorial activity of TFs with other cofactors, progressive priming of regulatory elements and the chromatin state of the DNA binding sequence in terms of accessibility [10]. 

The human genome encodes over 2000 different TFs: the particular set of TFs expressed in each cell type accounts for specific cellular programs and literature data clearly demonstrate that only few TFs are able to individually reprogram cell states when ectopically expressed (reviewed in [11]).

TFs function as repressors at least as often as they function as activators. Indeed, their binding to DNA can inhibit transcription either by recruiting co-repressors, or by turning off or competitively blocking the binding sites of activators, since TFs binding sites often overlap one another [12,13]. 

In general, combinatorial functions of different TFs are essential to modulate gene transcription. In the context of chromatin, the cooperativity is essentially established at the level of DNA binding and can be enhanced by protein–protein interactions between TFs [14]. 

### 1.3. Deregulation of Transcription Factors in Cancer

Extensive genome and transcriptome sequencing highlighted that multiple cancer diseases are associated with deregulated expression of TFs. In particular, both mutations and altered expression of various TFs have long been known to promote or take part to tumorigenesis by sustaining the oncogenic program [15,16].

Many TFs are deregulated as the consequence of activating or inactivating mutations. Heterozygous missense mutations occurring in TF genes can affect specific protein domains, conferring gain of function or dominant negative (DN) activity.

In addition to cancer-associated mutations, RNA sequencing clearly showed that TFs have aberrant splicing patterns of expression in cancer cells. Several reviews have recently provided a comprehensive background on the role of AS in human tumors and highlighted how cancer cells take advantage of misregulated RNA splicing to develop, proliferate and progress into therapy-resistant tumors [4,17,18,19,20].

Expression patterns of splice variants in normal versus cancer cells clearly indicates that: (i) the expression of TF isoforms is similar in multiple tumor types, suggesting that they could play a role in cancer cell biology, in some cases being even considered molecular markers [21] and (ii) for the most part, the effect of aberrant splicing of TFs affects global cellular processes, with considerable impact on cell proliferation, differentiation and cell death. 

Quantification of the relative proportions of TF isoforms in cancer tissues can be more informative than total TF amount when AS variants differently participate to cancer-associated processes. Here, we specifically and comprehensively addressed how splicing events of TF-encoding genes can produce proteins with transcriptional activity that oppositely influences cancer cell phenotypes. We focused on TFs with clinical correlation data, whose variants might have implications for diagnostic or prognostic or patients stratification purposes. 

We divided TFs into three categories: the first one is represented by TFs whose splice variants affect the cancer cell phenotype by regulating different classes of genes (NF-YA, STAT3, TCF4 and WT1). The second group is composed by TFs whose splice isoforms activate/repress the same target genes with different efficiency or even opposite effect (MEF2C, MITF and NFAT2). The third group encompasses TFs producing DN isoforms that compete with physiological TF activity (C/EBPβ, ERα, HELIOS, IKAROS, LEDGF, REST, SMAD4, TEAD 4 and KLF6).

## 2. TFs Splice Variants that Directly Control Different Transcriptional Programs

The balance between cell proliferation, differentiation and cell death programs has a key role in tumor development. Cancer cells show uncontrolled proliferation and the more aggressive tumors are associated to undifferentiated phenotypes. These processes are regulated by the activation of specific transcriptional programs that can be oppositely activated by isoforms of the same TF (Figure 1). Known alternatively spliced TFs whose isoforms directly regulate specific transcriptional programs, leading to opposite cell fates, have been summarized at the end of the paragraph in Table 1.

In detail, the activity of NF-YA, STAT3, TCF4 and WT1 isoforms in cancers has been documented and proposed for cancer molecular stratification.

### 2.1. Nuclear Transcription Factor Y (NF-Y)

The pioneer nuclear transcription factor Y (NF-Y) is a heterotrimeric complex composed of three subunits, NF-YA, NF-YB and NF-YC. NF-YA confers to the heterotrimer the ability to specifically recognize and bind the CCAAT box, a DNA regulatory element frequently found in regulatory regions of cell proliferation, cell death and key metabolic genes [22]. NF-Y coassociates with growth-controlling and oncogenic TFs, consistently with over-representation of the CCAAT box within promoters of genes differentially expressed in multiple cancer types [23]. The *Nf-ya* gene encodes for two splice variants, NF-YAs and NF-YAl, different in 28 amino acids within the N-terminal transactivation domain (TAD). Only recently, the two isoforms have been shown to affect activation of specific sets of genes. In embryonic stem cells and muscle cells, NF-YAs supports cell proliferation, in opposition to NF-YAl, which correlates with induced differentiation [24,25]. Forced expression of NF-YAs or NF-YAl DN isoforms show different behaviors on the expression of several stem cells genes, with only DN NF-YAs severely affecting *Klf4, Klf5, Arid1a, Fgf4, Sall4* and *Jarid2*. Moreover, the overexpression of NF-YAs is not able to activate bona fide NF-Y targets active after differentiation [24]. In muscle cells, the overexpression of NF-YAs or NF-YAl triggers opposite transcriptional and phenotypic effects: while NF-YAl enhances the differentiation program and directly activates the transcription of *Cdkn1c* (p57), *Mef2D* and *Six4* genes, NF-YAs blocks myotubes formation and preserves the activation of cell cycle genes, such as *CcnB1* [25]. These data support the hypothesis that the two NF-YA isoforms are not interchangeable and control different set of genes within the large NF-Y regulome. 

In endometrial tissues, the exclusive presence of NF-YAl observed in benign samples suggested that it may represent a marker of differentiation, in opposition to NF-YAs, which is expressed into cancer tissues and linked to poorly differentiated cells [26]. In breast and lung cancers, the ratio between NF-YAl/NF-YAs isoforms is dramatically shifted towards NF-YAs [27,28]. Despite this, partitioning of breast tumors according to NF-YAl/NF-YAs ratio highlighted a unique category with a high NF-YAl/NF-YAs ratio, that is NF-YAl^high^/Claudin^low^ subclass, composed by more aggressive tumors prone to metastasize [27]. NF-YAl^high^ has been suggested to be involved in increased expression of mesenchymal genes either indirectly or through direct control of specific epithelial-to-mesenchymal (EMT) TFs [27].

In lung squamous cell carcinomas (LUSC), the majority of patient tissues show a remarkable increase in NF-YAs and distinct gene signatures can be observed on the basis of the NF-YAs/NF-YAl ratio. NF-YAs^high^ tumors are enriched in metabolism and cell-cycle up-regulated gene categories. In opposition, these genes are down-regulated in NF-YAl^high^ tumors, characterized by the up-regulation of a promigration signature [28]. While NF-YAs directly activates cell cycle and metabolic genes, NF-YAl indirectly affects promigration genes. Additionally, in lung squamous adenocarcinomas (LUAD), specific NF-YAs increase and NF-YAl decreases is observed in all subtypes, TRU (terminal respiratory unit, bronchioid), proximal-proliferative (PP, magnoid) and proximal-inflammatory (PI, squamoid) [29]. 

### 2.2. Signal Transducer and Activator of Transcription 3 (STAT3)

STAT3 mediates signal transduction downstream of various cytokines, hormones, growth factors and interferons. It is ubiquitously expressed and is activated through phosphorylation of cytoplasmic monomers that, following dimerization, translocate to the nucleus and directly bind to the TTCC(G=C)GGAA consensus sequence. It activates different sets of genes depending on cell types and conditions. More than 70% of primary human tumors show STAT3 constitutive activation and many oncogenic pathways are activated by persistent STAT3 signaling [30]. The most commonly expressed isoform of STAT3 is the full-length STAT3α, a protein of 88 kDa predicted mass. The 83 kDa truncated isoform STAT3β is produced by AS of a conserved acceptor site in exon 23, causing a frameshift that introduces seven new amino acids and a stop codon in place of the C-terminal TAD. Originally tagged as DN, STAT3β possesses regulatory and transcriptional functions different from STAT3α [31,32]. The C-terminal amino acid tail of STAT3β prolongs nuclear retention of STAT3β homodimers compared to STAT3α, and increases homodimers stability and DNA-binding activity [33]. In multiple cancers, among which colon, lung, pancreatic, prostate, breast cancers and squamous cell carcinomas, melanoma and glioma, protein kinase C ε (PKCε) phosphorylates Ser727 exclusively present in STAT3α, thus increasing its DNA binding and transcriptional oncogenic activity [34]. In endometrial carcinoma, glioma, medulloblastoma, ovarian cancer and acute myeloid leukemia (AML), STAT3α enhances cell survival, proliferation and migration, promotes angiogenesis and metastasis. Moreover, it induces chemo-resistance by direct activation of target genes, such as *Survivin, Bcl-xl, Vegf, Cdkn1c* and *CyclinD1* [35,36,37,38,39]. 

STAT3β not only inhibits the constitutive activation of STAT3α, but directly controls the transcription of specific genes: in human melanoma xenografts, STAT3β-dependent upregulation of TRAIL receptor 2 induces cell apoptosis and consequently reduces tumor growth [40]. It can also activate *Fas* gene expression and therefore triggers apoptosis in cancer cells [41,42]. 

In AML patients, higher STAT3β/α isoforms ratio correlates with favorable clinical prognosis and prolonged survival while lower ratio is associated with higher overall mortality [43]. In blasts from STAT3β transgenic mice, STAT3β upregulates genes of the IL6/JAK/STAT3 signaling pathway and the reactome pathway for cell surface interactions at the vascular wall. Among up-regulated genes, *Cd62l* gene encoding for the SELL surface marker caught the attention, because it is expressed in newly diagnosed AML patients and its over-expression correlates with favorable prognosis. Other specific new identified STAT3β target genes, such as *Itgax* and *Cd177*, have been correlated to superior event-free survival [43].

### 2.3. T Cell Factor 4 (TCF4)

TCF4, also known as TCF7L2 (Transcription Factor 7 Like 2), is a member of the HMG-box-containing protein family and regulates gene transcription by binding the DNA motif (A/T)(A/T)CAAAG. TCF4 can act as either a transcriptional activator or repressor based on interacting proteins. Its activating function depends on direct interaction with nuclear β-catenin, while the repressive function is mediated by Groucho co-repressor binding in the absence of β-catenin [44]. TCF4 plays an important role in the regulation of growth and developmental processes, cell proliferation and apoptosis, inflammatory diseases, through the activation of the Wnt/β-catenin signaling cascade.

The *Tcf7l2* gene consists of 17 exons that are alternatively spliced into transcript variants with activity and detected in various cancer types, including colorectal cancer, brain, lung, liver and renal carcinoma [45,46,47,48]. Extensive splicing combinations at the C-terminal tail (exons 13-16) generate the isoforms TCF4E, TCF4M and TCF4S. Only TCF4E includes the complete C-clamp, a sequence-specific DNA binding domain, and two binding motifs for the transcriptional repressor CtBP [44,49]. TCF4E has the ability to antagonize β-catenin and p300 and contributes to the suppression of tumorigenesis by its unique transactivation properties at the Axin2 and Cdx1 genes, negative regulators of the Wnt/β-catenin pathway [44]. The absence or presence of the SxxSS amino acid motif by AS at the beginning of exon 9 results in two isoforms, TCF4J and TCF4K, characterized by different transcriptional activity. TCF4J works as tumor promoter by regulating the Wnt pathway [45]. TCF4J, which lacks of five amino acids and has low transcriptional activity, is able to activate cell transformation by enhancing cell proliferation, migration and colony formation. Compared to normal liver, in hepatocellular carcinoma (HCC) TCF4J expression is upregulated in opposition to TCF4K. TCF4J gives a higher proliferation rate and a survival advantage under hypoxic conditions with respect to TCF4K overexpressing HCC. In opposition, TCF4K functions as tumor suppressor [46,50]. Additionally, in lung cancer cells, TCF4K has shown the ability to inhibit cell proliferation and migration, presumably by the downregulation of *c-myc* and migration-relevant target genes [46].

Differently from the above described isoforms that regulate specific transcriptional programs, skipping of the splice donor at the 3’-end of exon 8 gives rise to TCF4N isoform that lacks the HMG-box DBD. TCF4N functions as DN factor of Wnt signaling by diverting β-catenin from target genes. TCF4N inhibits co-activation by β-catenin of TCF4-dependent promoters and redirects β-catenin to other TFs, such as SF-1 and C/EBPα, to transactivate TCF4-independent promoters [43,51]. TCF4N expression is down-regulated in esophageal cancer tissues, where Wnt/β-catenin signaling is activated [47].

### 2.4. Wilm’s Tumor 1 (WT1)

WT1 has a key role in normal cellular development and cell survival. It is a potent transcriptional regulator able to activate or repress the transcription of many target genes involved in the cell cycle, proliferation, differentiation, metabolism and apoptosis [52]. WT1 possesses a high affinity for the specific 10-bp DNA motif GCGTGGGAGT. Primarily expressed in the urogenital system in developing embryos, in the adult WT1 is found in kidney, the central nervous system and in tissues involved in hematopoiesis [53].

The *Wt1* gene undergoes two AS events: one on exon 5 encoding for 17 AA within the proline and glutamine-rich transregulatory domain, the other on exon 9 that encodes for lysine–threonine–serine amino acids (KTS) between zinc finger DBD 3 and 4. Consequently, four different isoforms (−17AA/−KTS, +17AA/−KTS, −17AA/+KTS and +17AA/+KTS) with distinct functions in various tumors are generated. ChIP-ChIP and ChIP-seq identified slightly different genomic binding sites between WT1 isoforms. Consistently, the four isoforms seem to control different set of genes associated to both tumor suppressive and oncogenic processes. 

In particular, the presence of both the +KTS and -KTS isoforms is essential for normal urogenital development. +KTS proteins have weak DNA-binding ability and can also bind RNA and the splicing machinery, playing a role in post-transcriptional regulation. -KTS isoforms bind mainly at promoters and enhancers of different set of genes, with a cell type- and developmental stage-dependent activity [54].

About 90% of Wilms’ tumor (WT), the most frequent kidney cancer in children, shows alteration of WT1 transcription. In particular, a disruption of exon 5 splicing decreasing the amount of the +17AA isoform is the most frequent alteration occurring in 56% of WT. This altered +17AA/-17AA isoforms ratio is consistent with the undifferentiated state of WT cells, since exon 5 inclusion increases during kidney maturation [55]. Tumor profilings allowed us to identify a set of genes specifically dysregulated in tumors with decreased +17AA/-17AA ratio. In detail, up-regulation of pro-proliferative genes, among which *Ccnd2, Pcna, N-Myc, E2f3* and *Top2A* and down-regulation of *Vegf, Igfbp5, Timp3, Arhb, c-Fos* and *Cd9* genes were observed. Moreover, since the few tumors with a high +17AA/-17AA ratio correlated with high VEGF expression and poor prognosis, WT1/VEGF expression levels has been proposed to define distinct WT subtypes [56]. 

The analysis of WT1 isoforms in other types of cancers highlighted that different cellular contexts can determine their transcriptional function.

Imbalances in +17AA/-17AA ratio seem to be associated to the development of hematological diseases [57]. In normal bone marrow, WT1 is expressed in progenitor cells and down-regulated during differentiation, while it is known to be overexpressed in leukemia cells. In acute myeloid leukemia (AML), WT1 could be a potential prognostic factor, marker for minimal residual disease and target of vaccination immunotherapy. In particular, +17AA variants are highly expressed at diagnosis and at relapse [58]. 

In mammary epithelial cells, WT1 -17AA/-KTS isoform acts as a tumor suppressor by inhibiting proliferation, inducing a G2 phase arrest of the cell cycle and promoting an acinar growth pattern. This seems to be triggered by direct *Cdkn1a* (p21) transcriptional upregulation. In contrast, +17AA/+KTS acts as an oncogene: its forced expression induces the cells to assume a mesenchymal morphology, causes *vimentin* overexpression and decreases the cell membrane expression of the *E-cadherin* target gene [59]. These different functions of WT1 isoforms are consistent with their expression pattern observed in breast clinical samples [60].

## 3. TFs Splice Variants that Differently Control the Same set of Genes

The second mechanism through which TF variants can perturb the balance between cellular processes envisages their ability to transcriptionally activate/repress target genes with diverse efficiency, due to different binding affinity for DNA or transcriptional co-regulators (Figure 2). These isoforms can not be classified as DNs since they preserve functional TAD and DBD. This is the case of the TFs MEF2C, NFAT and TCF4, whose isoforms do not have equivalent transactivating ability even if recruited onto the same chromatin region. 

TFs that belong to this category were recapitulated in Table 2, at the end of this section.

### 3.1. Myocyte Enhancer Factor-2C (MEF2C)

MEF2 family members, MEF2A, MEF2B, MEF2C and MEF2D, are known regulators of developmental programs, in particular myogenesis [61]. MEF2 proteins contain N-terminus highly conserved MADS (MCM1, AGAMOUS, DEFICIENS and SRF) and MEF2 domains, which are essential for DNA binding, dimerization and interaction with co-factors. Genetic alterations and consequent altered expression have been observed among MEF2 members in different cancer types, such as breast cancer, hepatocarcinoma, gastric cancer, AML, colorectal cancer and others [62]. In addition to altered expression levels, a switch in the expression of MEF2 spliced variants has been described in cancer, particularly for MEF2C. The two major splicing variants of MEF2C arise from mutually exclusive exons α1/α2, which generate ubiquitously expressed (MEF2Cα1) and muscle-specific (MEF2Cα2) isoforms. In muscle, they are both expressed but differently regulate proliferation and differentiation. The function of the peptide encoded by exon α, located adjacent to the N-terminal MEF2 DBD, is still unknown, although its inclusion seems to reduce MEF2C transcriptional activity [63]. The α1 isoform is associated to myoblast proliferation, while the muscle-specific α2 variant drives the late stages of myogenesis, similarly to Mef2Dα1 and Mef2Dα2 transcripts [64,65,66]. In rhabdomyosarcomas (RMS), the most common soft tissue tumor in the pediatric population, the ratio of α2/α1 is dramatically decreased compared to normal myoblasts, as the consequence of reduced MEF2Cα2 expression [65]. The mechanism at the basis of α2-specific induced myogenesis is imputed to differential phosphorylation of the two exons, which impacts on the association with histone deacetylases (HDACs). Indeed, exclusive phosphorylation of MEF2Cα1 allows its interaction with HDAC5 and induces HDAC4 and HDAC5 recruitment to regulatory regions of MEF2C muscle-specific target genes, which are therefore repressed. Forced expression of Mef2Cα2 is sufficient to reduce HDACs binding to target promoters, such as *Lmod2, Acta1, Tnnt1* and *Cdkn1A*, and hence triggers RMS differentiation and myotube formation [65]. 

### 3.2. Melanocyte Transcription Factor (MITF)

MITF, a basic helix–loop–helix–leucine zipper TF, plays a key role in differentiation, growth and survival of cells of the melanocytic lineage [67]. *Mitf* gene is transcribed from multiple promoters giving rise to multiple isoforms differing at their N-termini with diverse biological activities. The M-promoter is prominently used in the melanocyte lineage and responds to melanogenic extracellular signals, activating signal transduction mediated by tyrosine kinase and G-coupled receptors. MITF binds extended canonical E-box motif known as the M-Box (TCATGTGCT). In melanocytes, MITF AS might orchestrate the expression of positive and negative cell cycle regulators, thus controlling the balance between proliferation and differentiation [68]. Indeed, differential expression of MITF variants appears to influence phenotype, tumor biology and growth characteristics, allowing the identification of subgroups with different prognosis within melanoma patients [69]. The use of the differential splice acceptor in exon 6 in MITF mRNA gives rise to two isoforms: the (+) isoform includes exon 6a encoding the hexapeptide ACIFPT upstream of basic DBD, which lacks in the (−) isoform. The different function between the two isoforms might depend not only on direct DNA binding domain, with a slightly higher affinity for (+) than (−) MITF, but also on yet uncharacterized intra- or intermolecular interactions based on N-terminal domain and exon 6a-encoded hexapeptide [68,70]. (+) MITF antiproliferative activity is triggered by direct transcriptional activation of *Cdkn1A* (p21) and *Cdkn2A* (p16^INK4B^) expression and consequent hypophosphorylation of RB1 [71,72]. While (+) MITF is the predominant form expressed in normal epidermis and melanocytes, (−) MITF is highly expressed in melanoma cells and cutaneous metastasis, where it shows pro-proliferative activity [73].

### 3.3. Nuclear Factor of Activated T Cells 1 (NFAT2/ NFATC1)

NFAT family is composed by five proteins, four of which (NFAT1-4) are regulated by calcium signaling. NFAT binding motif is represented by 9-bp element including the consensus nucleotide sequence (A/T)GGAAA. NFAT proteins have an important role in immune response and in transcriptional regulation of genes encoding for signaling proteins, cell surface receptors, cytokines and regulators of cellular proliferation (e.g., p21WAF1/Cip1, CDK4, c-MYC and cyclin A2), cell differentiation (e.g., cytokines and growth factors) and apoptosis (e.g., FasL) [74,75,76,77]. 

Specifically, NFAT2 works as promoter of cell proliferation, repressor of cell death and inducer of cell transformation in melanoma, metastatic colorectal and pancreatic cancers [78,79,80,81]. Generation of NFAT2 spliced variants depends on alternative first exons that provide two different TADs. The NFAT2α and NFAT2β proteins differ only at the N terminus, with NFAT2α containing exon 1α-encoded 42 amino acids and NFAT2β having 28 differential amino acids encoded by exon 1β. NFAT2α regulates genes involved in cell survival and proliferation (*cyclinA2, cyclinD1* and *c-myc*), while NFAT2β takes part to the cell death program in various cell types suggesting a relevant proapoptotic role [78]. NFAT2α and NFAT2β could compete for the transcription of cell death genes to prevent or promote the apoptosis of activated T or B cells [78]. Indeed, NFAT2β may induce the up-regulation apoptotic genes, such as *Tnf-α* and *FasL*, more potently than NFAT2α through its conserved N-terminal TAD [78,82]. Burkitt lymphoma patients show high levels of NFAT2α associated to low NFAT2β levels, in opposition to what was observed in peripheral blood mononuclear cells from health donors. These data corroborated a pro-oncogenic role of NFAT2α in Burkitt lymphomas [82].

## 4. TFs Splice Variants with Dominant Negative Activity that Hamper TF Physiological Function

The last group is represented by splice variants that function as DN isoforms. DNs can act through different mechanisms, which consist of (i) cellular mislocalization, (ii) inability to bind target genes and even titration of the functional TF and iii) formation of transcriptional inactive complexes recruited on DNA.

Known TFs with DN splice variants that have been linked to different cancer-related processes were recapitulated in Table 3, at the end of this section.

### 4.1. DNs with Cellular Mislocalization

Mislocalized DNs generally have cytoplasmic compartmentalization because of the loss of the nuclear localization signal (NLS) that allows their nuclear import. Shifting the balance of splice variants in favor of cytoplasmic DNs affects the transactivation of transcriptional programs and can induce the malignant phenotype of cancer cells (Figure 3). Cytoplasmic DN isoforms can also sequester and inhibit the function of the endogenous active TF. Among TFs falling in this category, we will describe HELIOS and KLF6.

#### 4.1.1. Ikaros Family Zinc Finger Protein HELIOS

Ikaros represents a zinc-finger protein family mainly involved in lymphocyte development through the regulation of a wide range of processes, including proliferation and differentiation. Failure in their proper regulation is associated to cancer and autoimmunity. The Ikaros member HELIOS, encoded by the *Ikzf2* gene, is expressed in the T-cell lineage from the early stages of development and binds the GGGAA core motif to transcriptionally regulate genes involved in cell differentiation. Its deregulation has been observed in some patients with T-cell malignancies [83]. Three HELIOS DN isoforms, namely -V1, -V2 and -V3, have been described in human leukemic T-cell lines. HELIOS-V1 protein lacks exon 6 for the NLS and therefore has a cytoplasmic localization. HELIOS-V2 and -V3 lack exons 3–6, as well as all N-terminal zinc-finger motifs, thereby a functional DBD is absent [84]. Leukemia-type short HELIOS variants contribute to T-cell growth and survival. Indeed, the expression of HELIOS AS isoforms triggers deregulation of various downstream HELIOS target genes in T-cells compared to full-length protein. Among deregulated targets, genes regulating hematopoietic lineage commitment and differentiation (*Runx3, Irf8, Ebf3, Ptch1, Id2, C/Ebp zeta* and *Gata2*), hematopoietic-lineage markers (*Cd7, Cd24, Cd40LG, Cd34* and others) and leukemic translocation (*Mllt4* and *Tcf3* fusion partner) have been retrieved [84].

#### 4.1.2. Krüppel-like Factor 6 (KLF6)

The Krüppel-like zinc finger KLF6 is highly expressed in the placenta, spleen, thymus, prostate, testis, small intestine and colon [85]. KLF6 recognizes GC box or CACC-box DNA motifs in target promoters. It regulates cell growth, differentiation and adhesion and it is involved in multiple pathways, including adipogenesis, hematopoiesis, fibrotic response and repair during liver injury [85,86,87]. The full-length KLF6 acts as a tumor suppressor in a variety of cancers, including prostate, hepatocellular, colorectal, gastric, lung, head and neck, ovarian and brain cancers. This growth suppressive ability has been associated with many critical cancer-related pathways, including p53-independent upregulation of *Cdkn1a* (p21) [88,89], interference with the formation of cyclin D1/cdk4 complexes, *c-jun* and *E-cadherin* inhibition, *c-myc* activation, induction of apoptosis and cellular senescence [87,90,91]. The *Klf6* gene encodes for distinct biologically active proteins, the full-length KLF6 and the three alternatively spliced forms KLF6-SV1, -SV2 and -SV3, in both normal and cancerous tissues [92]. Deregulation of AS program occurs in prostate and ovarian cancer, glioblastoma, head and neck squamous cell carcinoma (HNSCC) and lung adenocarcinoma [88,93]. In particular, KLF6-SV1 has recently emerged as a tumor-promoting splice variant. It lacks all three zinc finger DBDs, leading to the expression of a mislocalized cytoplasmic isoform, which has opposite effects on cell proliferation, invasion, apoptosis and tumor dissemination compared to KLF6 [94,95,96]. KLF6-SV1 increased levels were recently associated with prognostic and predictive outcomes in multiple types of solid tumors. In breast cancer, KLF6-SV1 is an early driver of disease progression and independent prognostic risk factor for poor metastasis-free survival in a large cohort of early-stage breast cancer patients. KLF6-SV1 drives an EMT-like transition and triggers aggressive multiorgan metastatic disease in vivo. KLF6-SV1 expression could be used as a clinical marker for discriminating between indolent and lethal early-stage disease, and as a potential therapeutic target for invasive breast cancer [97]. In prostate cancer (PCa), the overall expression of KLF6 decreases with tumor progression in opposition to KLF6-SV1, which significantly increases in hormone-refractory metastases [98]. Similarly, increased KLF6-SV1/KLF6 ratio has been associated with aggressive clinical behavior HCC. KLF6-SV1induction is associated with HGF signaling, which induces c-MET and PI3K/AKT transduction and eventually affects the expression of SRSF3 and SRSF1 (Serine and Arginine Rich Splicing Factor 3 and 1), necessary for the expression of full length KLF6. Consistently, c-MET activation, SRSF3 downregulation and KLF6-SV1 expression have been proposed to identify subgroups of HCC patients with specific targetable alterations [99]. Besides, increased KLF6-SV1 expression and decreased KLF6 expression seem to play an important role in the regulation of cisplatin sensitivity in ovarian cancer [100].

### 4.2. DNs Impaired in DNA Binding Ability

The second mechanism at the basis of DN effects envisages AS isoforms that lack the DBD or other domains that are required for TF binding to DNA (Figure 4). An example is represented by AP-2α, a TF that regulates the expression of genes controlling proliferation, adhesion and invasion/angiogenesis in human melanoma [101,102]. The AP-2B isoform produced by AS lacks part of the DBD and the dimerization domain, which is required for DNA binding. AP-2B is an effective inhibitor of the transactivation potential of AP-2α, preventing its interaction with DNA. AP-2B overexpression in melanoma and teratocarcinoma cell lines leads to increased tumorigenicity, anchorage-independent growth, invasiveness and angiogenesis in vivo [103,104].

Below, we will focus on IKAROS and TEAD4 as TFs well representing how AS variants devoid of a functional DBD impact on cancer hallmarks. 

#### 4.2.1. Ikaros Family Zinc Finger Protein 1 (IKAROS)

IKAROS, encoded by the *Ikzf1* gene, plays a central role in hematopoiesis, influencing the self-renewal of hematopoietic stem cells, myelopoiesis and lymphopoiesis. It functions as a tumor suppressor in T and B lymphocytes. IKAROS controls gene transcription through the interaction with HDACs and chromatin-remodeling complexes, as Mi-2/NuRD and SWI/SNF, at canonical GGAAA binding sequences and pericentromeric heterochromatin alterations in the *Ikzf1* gene was shown in 83.7% of the patients affected by Philadelphia chromosome-positive (Ph+) acute lymphoblastic leukemia (ALL), consisting of haplo-insufficiency or a complete loss of expression, or in the presence of DN isoforms. The *Ikzf1* gene encodes multiple isoforms with different number of N-terminal zinc fingers and hence DNA binding affinity. Long IKAROS isoforms (IK1, IK2, IK3 and IKX) have at least three zinc fingers and are the functional isoforms. Shorter isoforms (IK4 to 10) cannot bind DNA and impair DNA-binding activity of longer IKAROS variants and other Ikaros family proteins, such as AIOLOS and HELIOS. Genome wide analysis identified many downstream IKAROS genes important for lymphocyte development, among which *Dntt* and *Rag* for VDJ recombination, *CD8α*, *CD3δ*, *Il2, Ahr, Runx1* and *Stat4* for T cell differentiation, and *cMyc* for B cell differentiation [105]. Hence, it is not surprising that reduction of IKAROS activity in the regulation of hematopoietic cell proliferation, survival and differentiation is induced by short DN isoforms and correlates to development of blast crisis in patients with chronic myeloid leukemia, myeloproliferative neoplasms, infant ALL, B-ALL and T-ALL [105,106,107,108].

#### 4.2.2. TEA Domain Family Member 4 (TEAD4)

TEAD4 is a member of the TEAD family (TEAD1-4) that binds the consensus muscle-CAT element (CATTCCA/T) through the N-terminal TEA domain, an evolutionarily conserved DBD. The C-terminal YAP-binding domain (YBD) allows direct interaction with transcription regulatory proteins [109]. TEAD proteins regulate the transcription of cell proliferation, differentiation and apoptosis genes. Their activation promotes tumorigenesis and progression through the overexpression of *Cyr61, Birc5, Ankrd1*, *vimentin* and *N-cadherin* genes in various tumors [109,110,111,112,113].

TEAD4 works downstream of the Hippo-YAP signaling, a key regulatory pathway in cell development and homeostasis that controls cell proliferation, cell contact inhibition and organ size, frequently dysregulated during tumorigenesis [109]. The TEAD4 short (TEAD4-S) isoform that lacks the N-terminal DBD but maintains intact YBD is produced by exon 3 skipping. Differently from the canonical TEAD4-FL that is found exclusively in the nucleus, TEAD4-S is located in both the cytoplasm and nucleus, and works as a DN isoform of TEAD4-FL in the regulation of YAP activity. By inhibiting the YAP signaling, the short isoform is responsible for the dysregulation of cell cycle, cell proliferation and RNA processing pathways normally activated by YAP-TEAD4-FL interaction [113,114]. The two isoforms show the opposite roles in cancer in vitro and in vivo, with TEAD4-FL being a tumor promoter and TEAD4-S a tumor suppressor [115]. Clinical investigation and TCGA data analysis highlighted that TEAD4-S expression is commonly reduced in human cancers and patients with elevated TEAD4-S levels show improved survival [111,115]. In particular, in lung cancer cells, which have lower TEAD4-S level compared with normal cells, TEAD4-FL activates transcription of *N-cadherin* and *vimentin* genes inducing EMT of tumor cells, while TEAD4-S suppresses EMT markers [115]. RBM4, a splicing factor that functions as a tumor suppressor in many cancers, has been pointed out as a possible regulator of TEAD4 splicing of exon 3. Indeed, the activation of RBM4 increases TEAD4-S production and inhibits tumorigenesis [115].

### 4.3. DNs with Altered Regultory Ability

The third DNs category is composed by those TF isoforms that can still bind DNA at target genes but can not play their regulatory activity due to the lack of transactivating (TAD) or repressive (RD) domains or to the impaired ability to recruit onto chromatin other TFs or transcriptional co-regulators (Figure 5). Among these, unbalanced levels of ΔN isoforms of p53 family members (p53, p63 and p73), which lack the TAD and can not transcriptionally activate p53 target genes as TA splice variants, are frequently observed in tumors and are associated to cancer cell survival [116,117,118]. Another example is represented by the *Dmtf* gene, which physiologically controls the Arf-p53 pathway, through transactivation of the *Cdkn2a* (p14ARF) promoter and physical interaction with p53. The DMTF1β isoform that lacks the C-terminal TAD and the DNA binding ability of the full-length α isoform is overexpressed and associated with poor patient outcomes in breast cancer and contributes to human leukemogenesis [119,120,121].

In some cases, DNs with oncogenic activity are exclusively expressed in tumors, as miniSOX9, the DN SOX9 variant that lacks the TAD and is expressed only in colorectal cancer tissues [122]. In other cases, the DN isoform can function as oncosuppressor that dampens the pro-oncogenic variant. This is the case of ΔFOSB, which lacks the majority of the TAD of the FOS protein and alters the transcriptional activity of AP-1 complexes, thus inhibiting tumor-promoting pathways [123].

Below, we will better dissect C/EBPβ, LEDGF, REST, SMAD4 and ERα as examples of TFs with isoforms characterized by different transcriptional potential based on coactivators/corepressors chromatin recruitment.

#### 4.3.1. CCAAT-Enhancer Binding Protein β (C/EBPβ)

C/EBPβ is a member of the C/EBP family of leucine-zipper (bZIP) TFs that bind DNA as dimers and regulate the transcription of genes containing the T(TG)NNGNAA(TG) motif. C/EBPβ is expressed in many tissues including the liver, brain, intestine and skin. Recent studies reported that C/EBPβ has a role in cell cycle regulation, differentiation, apoptosis and senescence [124]. C/EBPβ is essential for mammary epithelial and granulosa cells differentiation, macrophages functioning and proinflammatory program in microglia, brown adipose tissue formation, microglia and neuronal apoptosis [125,126,127,128,129,130]. 

The single exon and intronless *Cebpb* gene is alternatively translated into three C/EBPβ isoforms, LAP1, LAP2 and LIP. LAP1 and LAP2 differ in 21 N-terminal amino acids as the result of internal translation initiation from the downstream LAP2 alternative translational start codon. Both isoforms are transcriptional activators, with LAP2 being a stronger transactivator than LAP1, presumably via regulation of C/EBPβ protein tertiary structure and unique N-terminal protein–protein interactions [131]. LIP isoform lacks the N-terminal TAD and most of the negative regulatory domain, thus functioning mainly as a DN regulator of transcription, although in some cellular contexts LIP can act as transcriptional activator by interacting with other cofactors [131]. The expression of C/EBPβ-LAP1 variant, but not LAP2 and LIP, is significantly downregulated in hepatocellular carcinoma, squamous cell carcinoma, breast cancer and elevated LIP:LAP ratio is observed in high proliferative tumors [132,133,134]. While the overexpression of LAP isoforms determines cell cycle arrest or differentiation, as shown in hepatocytes, hematopoietic cells and adipocytes, LIP induces proliferation and contributes to tumor cells survival. Increased LIP expression antagonizes LAP activity and the high LIP:LAP ratio favors *c-myc* activation and inactivation of cell cycle inhibitor genes [131,135]. LIP isoform contributes to cell survival and tumor progression, as demonstrated in hepatoma cells. LAP1/2 overexpression suppresses migration in vitro and metastasis in vivo, in addition to preventing tumor formation in hepatocellular carcinoma xenografts [132,136]. In breast cancer, both LAP and LIP are overexpressed, but the LIP isoform is more highly expressed in the most aggressive, poorly differentiated specimens [131].

#### 4.3.2. Lens Epithelium-Derived Growth Factor (LEDGF)

The human *Psip1* (PC4 and SFRS1 Interacting Protein 1) gene encodes for two transcripts generated through alternative splicing, LEDGF/p75 (exons 1–15) and LEDGF/p52 (exons 1–9 and part of intron 9). They have a common N-terminal portion, important for chromatin binding at active genes, which harbors a Pro-Trp-Trp-Pro (PWWP) domain (which recognizes H3K36-me2 and -me3) and a tripartite DNA-binding element composed of a NLS and two AT-hook sequences. Both isoforms function as transcription co-activators, but they display distinct nuclear patterns, owing to different binding partners and functions in the cell [137]. The C-terminus uniquely present in the LEDGF/p75 isoform contains the HIV-1 integrase binding domain and the region recognized by human anti-LEDGF/p75 autoantibodies, whereas LEDGF/p52 has eight distinctive intron-derived C-terminal amino acids [138,139,140]. As a transcriptional coactivator, LEDGF/p75 recognizes di- and tri-methyl K36 of histone H3, recruiting protein complexes at active chromatin [139,141], but it was also reported a sequence-specific binding to heat shock elements (HSE; nGAAn) and stress-related elements (STRE; T/AGGGG) [142]. The LEDGF/p75 protein is ubiquitously present in mammalian cells and it has been implicated in inflammatory and autoimmune conditions. Besides, it is a key cellular cofactor that drives HIV-1 integration into transcriptionally active sites in host chromatin and it has recently emerged as a tumor-associated antigen and a stress–survival oncoprotein [140,143,144]. LEDGF/p75 is upregulated in cancer cells, whereas LEDGF/p52 is expressed at relatively low levels, with a high p75/p52 ratio [140,145]. In cancer cells, LEDGF/p52 overexpression induces apoptosis and antagonizes the prosurvival transcriptional activity of LEDGF/p75, perhaps by competing for promoter regions or interactions with distinct transcription factors. p75 has been reported to interact with several proteins, among which the menin/MLL complex and MeCP2 epigenetic factors, and facilitates their association to chromatin [146,147]. 

In breast cancer, p75 binding to chromatin modulates the expression of cell cycle genes such as *CcnD1, Cdk4* and *Cdk6*, by regulating RNA pol II recruitment to promoter regions [148]. The positive correlation identified between p75 levels and basal-like subtype or triple negative breast cancers has a significant impact on patient survival. LEDGF/p75 expression is upregulated in prostate, colon, thyroid, liver, uterine, bladder and breast cancers, relative to corresponding normal tissues [143,149]. It participates in carcinogenesis and therapy resistance, through activation of a stress protective pathway and increasing tumor aggressive properties, such as high cell proliferation and survival, clonogenicity, migration and angiogenesis. LEDGF/p75 can be upregulated in chemoresistant cancer cells, as it stabilizes lysosomes and protects cancer cells against lysosomal cell death induced by anticancer agents [145,149]. The relative expression ratio of the prosurvival vs. proapoptotic LEDGF variants might determine tumor cell response in the presence of oxidative stress or chemotherapeutic drugs [140].

#### 4.3.3. RE1 Silencing Transcription Factor (REST)

REST, also known as neuron-restrictive silencing factor (NRSF) was originally identified as a transcriptional repressor of neuronal differentiation genes in non-neuronal cells and neural stem cells. Nevertheless, REST is actually recognized as an important transcriptional and epigenetic factor, which controls a multitude of genes, in neuronal and non-neuronal cells [150]. The N-terminal repression domain RD1 can recruit Sin3/HDACs complexes, while C-terminal RD2 can mediate the binding of other co-repressors, such as CoREST, HDACs, the histone H3K4 lysine demethylase LSD1, the H3K9 methyltransferases G9a and the SWI/SNF chromatin-remodeling complex. The DBD of REST/NRSF can recognize the broadly-distributed repressor element-1 (RE-1), whose core consensus motif is NT(T/C)AG(A/C)(A/G)CCNN(A/G)G(A/C)(G/S)AG. REST/NRSF aberrant expression or mutations were observed in a multitude of cancers. While it functions as a tumor suppressor in prostate, colon, breast and lung epithelial cells, where it protects from neuroendocrine transformation, REST/NRSF shows tumor-promoting functions in medulloblastoma and glioma, ovarian and pancreatic cancers [151,152,153,154,155,156]. The *Rest* gene can generate many different coding and non-coding transcripts through the usage of three alternative promoters and alternative exons. This leads to alterations in the protein structure and function and few isoforms were comprehensively described as DN proteins. REST1, which contains RD1 and zinc finger motifs 1–4, and REST-5FΔ, with an in-frame deletion of zinc finger motif 5, have a cytosolic localization, therefore belonging to the DNs category described in Section 4.1. REST4 is the most characterized protein isoform, whose over-expression has been associated to cancer onset and aggressiveness [155,157,158]. REST4 and REST4-like proteins that contain RD1 and zinc finger motifs 1-5, but not RD2, are generated by a group of variants, due to inclusion of alternative N-terminal exons of variable length that introduce a premature stop codon. These isoforms lose the transcriptional silencing ability of REST/NRSF, due to a lack of C-terminal co-repressor binding domains, and have a reduced DNA-binding function and hinder the wild type REST/NRSF activity, by inhibiting its binding to RE-1 motifs. Loss of REST/NRSF and increased REST4 splicing were identified in a subset of aggressive breast tumors and also positively correlated to development of castration-resistance prostate cancers with neuroendocrine phenotype [153,159]. Among REST4-like proteins with DN activity, REST-N62 and REST-N50 (or sNRSF) have been described in human neuroblastoma and lung tumors, respectively [156,160]. Specifically, the truncated REST-N50 variant, which results from the incorporation of an alternative N-terminal exon of 50-bp, is expressed at high levels in neuroendocrine-type small cell lung cancers (SCLCs) relative to non-small cell lung cancer or normal lung tissue [156].

#### 4.3.4. SMAD Family Member 4 (SMAD4)

The TF SMAD4 belongs to the SMAD family, a group of intracellular proteins that transduce extracellular TGF-β signals directly to the nucleus and regulate transcription. SMAD4, also called Co-SMAD, is the central mediator of the TGF-β and bone morphogenic protein (BMP) signaling pathways. The N-terminal MH1 is responsible for the DNA binding at level of the specific palindromic sequence GTCTAGAC and contains the NLS, while the C-terminal MH2 domain is important for heteromeric SMAD complexes aggregation, which is necessary for SMAD4 transcriptional activity [161,162]. The linker region between MH1 and MH2 contains the TAD and a nuclear export signal (NES).

SMAD4 is involved in numerous processes such as proliferation, differentiation and apoptosis, and many studies reported SMAD4 tumor suppressive activity. Indeed, SMAD4 expression is down-regulated in several types of cancers causing the alteration of TGF-β pathway and resulting in tumorigenicity, angiogenesis, invasion and resistance to chemotherapy [163,164,165]. SMAD4 inactivation can promote tumor progression and metastasis development in a tumor type-dependent manner. In vivo, SMAD4 deletion contributes to tumorigenesis in head and neck, pancreatic, colon and ovarian human cancers [162,166]. Oppositely, the over-expression of SMAD4 enhances apoptosis and inhibits cancer cell proliferation [162,167].

*Smad4* mRNA is alternatively spliced in the linker region to give six different isoforms, Δ3, Δ4, Δ5-6, Δ6, Δ4-6 and Δ4-7, all of which are able to form complexes onto DNA. Exons 5, 6 and 7 are required for SMAD4-mediated transcriptional activation induced by TGF-*β* signaling. TAD deletion that includes most of exon 6 and the entire exon 7 has a DN effect on TGF-*β* signaling, since these mutants are still able to form complexes with other SMAD proteins and TFs, but are not able to activate transcription. Differently, exon 3 and 4 encompass the NES and their deletion leads to improper nuclear accumulation of SMAD4 [168,169]. 

Differentiation of thyroid cells is controlled by TGF-β that is also the main negative regulator of thyrocyte proliferation. Δ4-7 Δ6, Δ4-6 and Δ5-6 isoforms that switch off the SMAD4 function have been associated with the papillary thyroid, neuroblastoma and breast cancers [168,169,170,171].

#### 4.3.5. Estrogen Receptor Alpha (ERα)

ERα is a ligand-activated TF that regulates gene transcription by binding specific DNA estrogen-response elements–EREs (AGGTCANNNTGACCT) as a homodimer or heterodimer with ERβ. The *Esr1* gene encodes a full-length ERα protein (or ERα-66) that contains a N-terminal ligand-independent activation domain (AF-1), a central DBD, a C-terminal ligand binding domain (LBD) and ligand-dependent activation function region (AF-2). NLS and NES domains allow proper subcellular distribution of ERα and its transcriptional activity. ERα enhances cell proliferation and survival, and exhibits anti-apoptotic and anti-inflammatory activities, functioning as either a transcriptional activator or repressor of target genes. ERα expression is reported in several types of cancer, particularly in hormone-sensitive tumors such as breast, ovarian, endometrial and prostate.

ERα is overexpressed in approximately 50–70% of breast cancers, where it promotes tumorigenesis and tumor progression. Different mRNA variants originate from alternative promoter usage and AS, and the ratio of wild-type vs. variant mRNAs can play a role in the estrogen response and antihormonal therapy sensitivity. These alternative isoforms are normally present together with the wt receptor and can exert dominant-positive or -negative effects in tumors.

ERα-46 lacks exon 1 encoding for the N-terminal TAD (AF-1), but it can dimerize with the wild-type ERα acting as a competitive inhibitor of ERα and suppressing breast cancer proliferation. ERα-46 is largely expressed in ERα-positive breast tumors, where the ERα-46/ERα-66 expression ratio inversely correlates with tumor size and grade [172]. The Δ3 and Δ7 isoforms, lacking exon 3 and 7, respectively, also act as DN splice variants, interfering with ERα-mediated transcriptional regulation. Substantial increase of the ERαΔ3 isoform that lacks part of the DBD seems to be an early event in breast carcinogenesis and its overexpression in cell lines determines enhanced anchorage-independent growth, clonogenic ability and in vitro invasion. ERαΔ7 protein lacks the AF-2 TAD domain and a portion of the hormone binding domain, it is not transcriptionally active and is a potent DN isoform for ERα. ERαΔ7 is the most frequently observed variant in breast cancer, regardless of ER status. In addition to breast cancer, it is highly expressed in meningiomas, endometrial hyperplasias and moderate- to well-differentiated endometrial adenocarcinomas (reviewed in [173]). ERα−36 is generated from an alternative first exon in *Esr1* gene and a different C-terminal exon, coding for a distinctive 27-amino acid sequence. ERα-36 contains DBD and LBD but not the N- and C- terminal TADs, thus lacking intrinsic transcriptional activity, but hindering the transactivation signaling of full-length ERα. This isoform is highly localized in the cytoplasm and at the plasma membrane, where it can mediate non-nuclear receptor functions that promote proliferation and aggressiveness of breast tumors. ERα-36 correlates with shorter disease-free survival, both in ERα-66-negative and -positive breast cancer patients, and with altered response to therapy, as tamoxifen. The oncogenic activity and prognostic value of ERα−36 has been demonstrated also in primary and secondary hepatocellular carcinoma [174].

Finally, the ERαΔ5 variant that retains AF-1 activity and DNA binding ability behaves as constitutively active TF: it promotes gene transcription even in the absence of hormone stimulation [173]. The expression of ERαΔ5 has been described in hepatocellular carcinoma where it could be used as classification predictors for survival [175].

## 5. Conclusions and Prospects

Cancer-associated genetic alterations can change the expression and activity of proteins participating in nearly all cellular processes, among which gene transcription. Aberrant activity of specific TFs occurs in most human cancer cells as the consequence of a change in expression level, stabilization/degradation modification or alteration in DNA binding and transactivating activity. Hence, specific inhibitors have been developed against oncogenic TFs associated with hallmarks of cancers, such as NFκB, MYC, STATs, FOXO, p53, HOXs and others (recently reviewed in [176,177]). Direct targeting of TFs, historically considered "undruggable", should have unique effects compared to more aspecific kinase inhibitors and is therefore a promising anticancer therapy. Some TFs inhibitors, currently in clinical development or in clinical trials, are able to disrupt the cancer transcriptional programs, for example through increased differentiation or cell death activation [176].

This encouraging scenario gets complicated because of AS events in TF-coding genes that can generate splice isoforms with different transcriptional activity. Indeed, oncogenic or oncosuppressive TFs can be encoded by the same gene. Consistently, molecular and bioinformatics investigations showed that AS events allow one to define molecular cancer subtypes and can predict patient outcome. In cancer samples, the balance of TF isoforms is often shifted towards the oncogenic variant, which in some cases is even expressed uniquely in cancer samples compared to the normal counterparts. Here, we focused on those TFs whose splice variants have been associated to transcriptional cancer signatures and therefore could be used for molecular stratification of cancer patients and as new selective targets for drug development (Figure 6).

The identification of TFs isoforms with ying–yang roles highlights the importance of deep analysis not only of the total TF expression levels but also of relative ratios between splice variants in normal vs. cancer tissues or in different cancer subgroups. The development of high-throughput technologies and computational tools highlighted cancer cells heterogeneity, in which alternative RNA splicing has a considerable role. Once identified, standard techniques, such as PCR-, microarray- or antibody-based applications, could be easily used for detection of splice variants also in clinics [178].

Another interesting aspect of AS is the association between gene isoforms and drug- or radiosensitivity. Genome-wide meta-analyses have been performed to identify isoform-based biomarkers predictive of drug response in vitro. Literature data describe the existence of a correlation between chemotherapeutic agents effectiveness and "non-TF" protein variants expression, such as the androgen receptor isoform AR-V7 and abiraterone/enzalutamide in patients with castration-resistant prostate cancer [179] or BRCA1-delta11q and PARP inhibitors/cisplatin in breast cancer [180]. As for TFs, deregulated ΔNp63 and ΔNp73 isoforms have been shown to contribute to chemo- and ionizing radiation-resistance in human cancers, such as cervical and colon cancers [181,182]. Therapeutic modulation of TAp63/ΔNp63, TAp73/ΔNp73 and mutant p53 levels has been proposed as targeted therapy in tumors characterized by p53 mutations and/or ΔNp63 or ΔNp73 overexpression [183].

Targeting specific splicing factors or their post-translational modifications, rather than cancer-associated spliced proteins, all represent an emerging therapeutic opportunity to personalize and maximize cancer treatment responses [17]. However, the most used strategy to target specific oncogenic isoforms is represented by antisense oligos (ASOs) or splicing-switching oligos (SSOs). The first one aims at affecting splicing events by synthetic ASOs that recognize either exon–intron junctions or regulatory sequences in introns or exons. The second one is represented by ASOs that alter the recognition of pre-mRNA splice sites by splicing factors. The most advanced SSO with demonstrated activity in cancer in vivo currently include STAT3 and other non transcriptional proteins, such as the apoptotic regulator BCL2L1, the signal transducers ERBB4 and the p53 regulator MDM4 [184].

In conclusion, transcriptional dependencies of cancer cells from specific TFs isoforms highlight their fundamental activity in controlling cell identity. Despite this key role, alterations in TFs splicing events have been often underestimated and only total expression levels of a specific TF has been investigated and correlated to clinicopathological features and prognosis. Mechanistic studies are required to pinpoint and understand how unbalanced TFs isoforms can account for the neoplastic state in order to open new opportunities for precision medicine in cancer.

## Figures and Tables

**Figure 1 cells-09-00760-f001:**
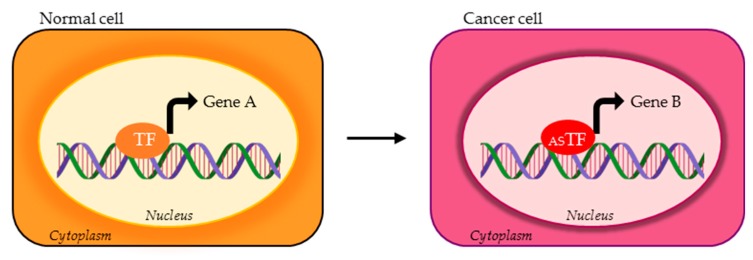
Possible activity of transcription factor (TF) alternative splicing (AS) isoforms in normal vs. cancer cells. The TF variant expressed in cancer cells (red ASTF) can bind and transcriptionally regulate different set of genes compared to the TF physiologically expressed in normal cells (orange TF). Gene A and gene B represent two different targets of TF isoforms.

**Figure 2 cells-09-00760-f002:**
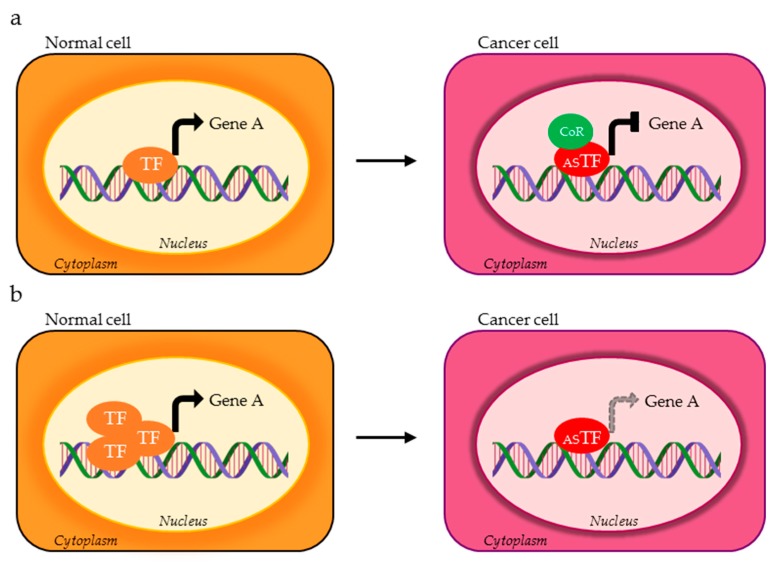
TF isoforms expressed in cancer cells (red ASTF) can (**a**) oppositely control the expression of the same target gene compared to the physiological TF variant (orange TF) through the recruitment of different co-regulators (green CoR) or (**b**) have reduced affinity for the target gene.

**Figure 3 cells-09-00760-f003:**
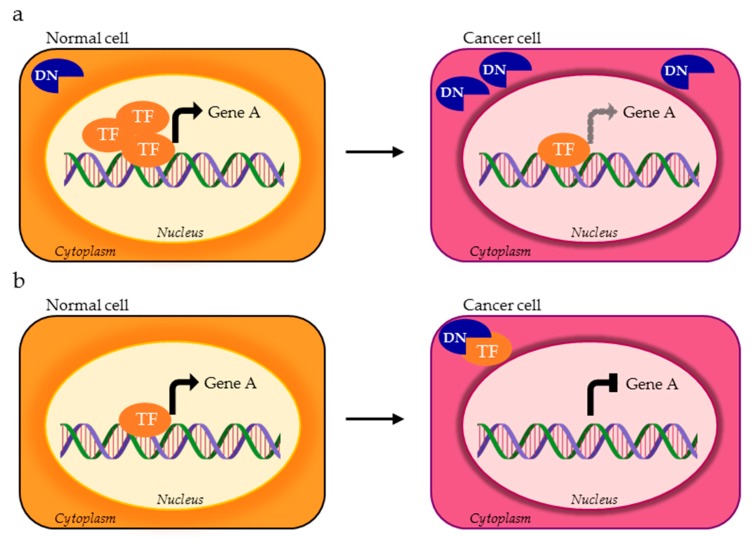
AS isoform of a TF can be mislocalized within cancer cell and exert a dominant negative (DN) activity as the consequence of (**a**) excessive expression of the non functional isoform (blue DN) compared to the functional one (orange TF) or (**b**) cytoplasmic titration of the functional TF variant.

**Figure 4 cells-09-00760-f004:**
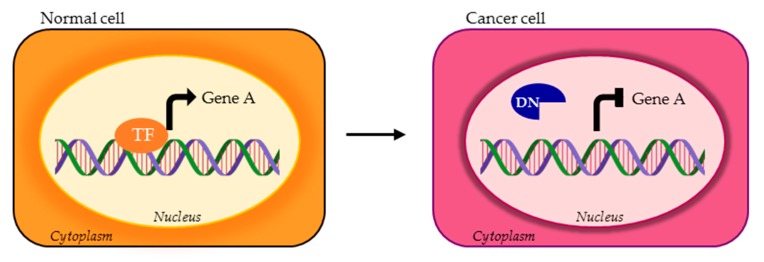
TF isoforms can lose the DBD or other domains required for DNA binding of the canonical TF (orange TF), consequently acting as DNs (blue DN).

**Figure 5 cells-09-00760-f005:**
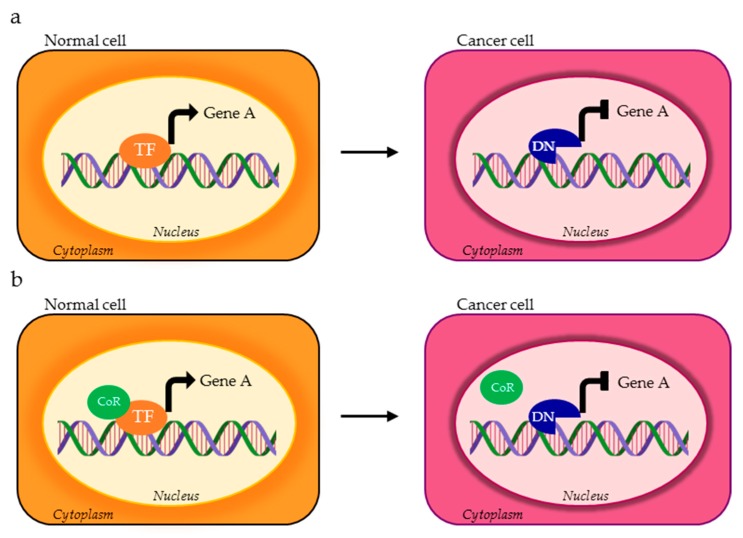
TF isoforms acting as DNs (blue DN) can preserve DNA binding to target genes but do not have transactivation or repressive potential because of (**a**) the loss of the TAD or repressive domain (RD) by AS or (**b**) impaired recruitment of transcriptional co-regulators (green CoR) necessary for proper transcriptional regulation.

**Figure 6 cells-09-00760-f006:**
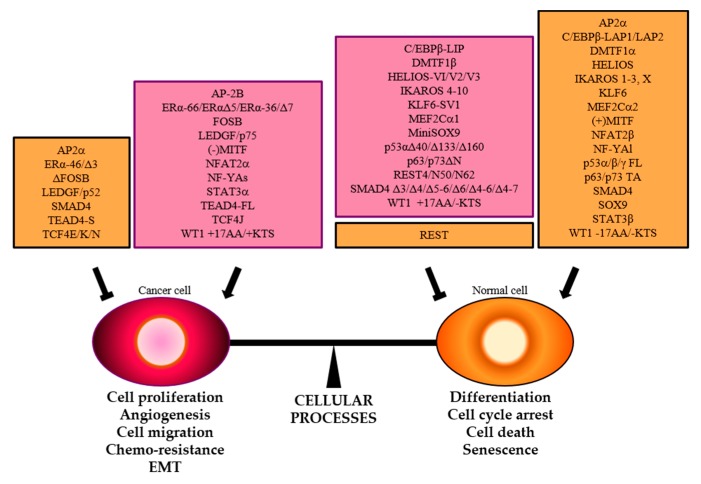
The balance between cellular processes that enhance or inhibit cancer hallmarks is controlled by TFs. Specific splice variants that promote (purple boxes) or hamper (orange boxes) neoplastic transformation are reported.

**Table 1 cells-09-00760-t001:** Transcription factors (TFs) and respective splice variants that promote tumor formation or growth through direct regulation of specific transcriptional programs. Tumor-promoting (TP) splice variants with clinical correlation data and relative AS domains have been listed. *These isoforms have tumor-type specific oncogenic activity. DBD (DNA binding domain), TAD (transactivation domain).

TF	DNA Binding Motif	Physiological Regulated Processes	Splice Variants	TP Splice Variants	AS Domains in TP Splice Variants	Ref.
**NF-YA**	CCAAT	Cell proliferation and differentiation, metabolism, cell death	NF-YAs NF-YAl	NF-YAs	TAD	[26,27,28,29]
**STAT3**	TTCC(G=C)GGAA	Cell proliferation, cell death	STAT3α STAT3β	STAT3α	TAD	[31,32,33,34,35,36,37,38,39,40,41,42,43]
**TCF4**	(A/T)(A/T)CAAAG	Cell proliferation, apoptosis	TCF4 TCF4N/E/M/B/S/K/J	TCF4J	SxxSS motif	[44,45,46,47,48,49,50,51]
**WT1**	GCGTGGGAGT	Cell proliferation and differentiation, metabolism, apoptosis	-17AA/-KTS -17AA/+KTS +17AA/-KTS +17AA/+KTS	-17AA* +17AA* +17AA/+KTS	TAD DBD	[54,55,56,57,58,59,60]

**Table 2 cells-09-00760-t002:** Transcription factors (TFs) and respective splice variants with oncogenic properties characterized by different activating/repressing ability on the same target genes physiologically regulated by isoforms expressed in normal cells. Tumor-promoting (TP) splice variants with clinical correlation data and relative AS domains have been listed. DBD (DNA binding domain), TAD (transactivation domain), Y (pyrimidine), R (purine).

TF	DNA Binding Motif	Physiological Regulated Processes	Splice Variants	TP Splice Variants	AS Domains in TP Splice Variants	Ref.
**MEF2C**	YTA(A/T)_4_TAR	Muscle cell proliferation and differentiation	Mef2Cα1 Mef2Cα2	Mef2Cα1	Adjacent to DBD	[65]
**MITF**	TCATGTGCT	Melanocyte proliferation and differentiation	(+) MITF (−) MITF	(−) MITF	Adjacent to DBD	[68,69,70,71,72,73]
**NFAT2**	(A/T)GGAAA	Cell proliferation and differentiation, apoptosis, inflammatory response	NFAT2α NFAT2β	NFAT2α	TAD	[78,82]

**Table 3 cells-09-00760-t003:** Transcription factors (TFs) and respective splice variants expressed in tumor cells that function as dominant negative and hamper TF physiological activity. Tumor-promoting (TP) splice variants with clinical correlation data and relative AS domains have been listed. DBD (DNA binding domain), TAD (transactivation domain), NLS (nuclear localization signal), IBD (integrase binding domain), CTD (C-terminal domain), Y (pyrimidine), R (purine), S (G or C), N (any base).

TF	DNA Binding Motif	Physiological Regulated Processes	Splice Variants	TP Splice Variants	AS Domains in TP Splice Variants	Ref.
**AP-2α**	GCCNNNGGC	Development, cell growth and differentiation, apoptosis	AP-2α AP-2B	AP-2B	Dimerization domain	[103,104]
**CEBP** **β**	T(TG)NNGNAA (TG)	Cell cycle, differentiation, apoptosis and senescence	LAP1 LAP2 LIP	LIP	TAD	[131,132,133,134,135,136]
**DMTF1**	CCCG(G/T)ATGT	Cell proliferation, apoptosis	DMTF1α/β/γ	DMTF1β	TAD DBD	[119,120,121]
**ER** **α**	AGGTCANNNTGACCT	Cell proliferation, apoptosis, inflammation	ERα (ERα-66) ERαΔ1 (ERα-46) ERαΔ2/Δ3/Δ4/ Δ5/Δ7 ERα−36 ERα−30	ERα-66 ERα−36 ERαΔ3/Δ5/Δ7	TAD DBD LBD	[172,173,174,175]
**FOSB**	TGAC/GTCA	Cell proliferation and differentiation, apoptosis, stress response	FOSB ΔFOSB	FOSB	TAD Degron domain	[123]
**HELIOS**	GGGAA	T-lineage differentiation	HELIOS HELIOS-V1/V2/V3	HELIOS-V1/V2/V3	NLS DBD	[84]
**IKAROS**	GGAAA	Hematopoiesis, myelopoiesis, lymphopoiesis	IK1-10 IKX	IK4-10	DBD	[105,106,107,108]
**KLF6**	GC box CACC box	Cell proliferation and differentiation, adhesion, tissue repair	KLF6 KLF6-SV1/SV2/SV3	KLF6-SV1	DBD	[88,92,93,94,95,96,97,98,99,100]
**LEDGF**	NGAAN T/AGGGG	Neuroepithelial stem cell differentiation and neurogenesis, stress-induced apoptosis, lens epithelial cell growth and differentiation, host-virus interaction	LEDGF/p52 LEDGF/p75	LEDGF/p75	IBD CTD	[140,143,144,145,146,147,148,149]
**p53/p63/p73**	RRRC(A/T)(A/T)GYYY	Cell cycle arrest, cell death, genome stability, cell differentiation, development	p53α/β/γ Δ40/Δ133/ Δ160 p53α/β/γ TA/ΔN p63 TA/ΔN p73	Δ40p53α Δ133p53α Δ160p53α ΔN p63/p73	TAD DBD	[116,117,118]
**REST/NRSF**	NT(T/C)AG(A/C)(A/G)CCNN(A/G)G(A/C)(G/S)AG	Cell differentiation	REST REST1/4/5 REST-N50/N62 REST-5FΔ	REST4 REST-N50 REST-N62	DBD NLS	[153,155,156,157,158,159,160]
**SMAD4**	GTCTAGAC	Cell proliferation and differentiation	SMAD4 SMAD4 Δ3/ Δ4/Δ5-6/Δ6/ Δ4-6/Δ4-7	SMAD4 Δ3/ Δ4/Δ5-6/ Δ6/Δ4-6/Δ4-7	Linker domain TAD	[168,169,170,171]
**SOX9**	(A/T)(A/T)CAA(A/T)G	Stem cell maintenance and commitment, differentiation, matrix deposition	SOX9 MiniSOX9	MiniSOX9	TAD	[122]
**TEAD4**	CATTCCA	Cell proliferation and differentiation, apoptosis	TEAD-FL TEAD-S	TEAD-FL	DBD	[111,113,114,115]

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
