# Peer review of "Transcription Factors in Cancer: When Alternative Splicing Determines Opposite Cell Fates"

_cells, 2020, doi:10.3390/cells9030760_

Round 1

Reviewer 1 Report

The review written by Belluti and Rigillo, is well written informative and timely. There are no major flaws therefore, I have gone into more detail on much more minor issues. Overall, the review is very long, and can read like a thesis rather than a review. I would recommend publishing this manuscript as long as the following are addressed to a satisfactory standard:

Minor points

Narrative:

The review suffers from a clear narrative. While the purpose is clearly understood (to review transcription factors which are alternatively spliced) there is no clear rationale for why the discussed transcription factors are being highlighted in opposed to others. It would be good to include this rationale in the final sentence of the introductory paragraph. If this list is exhaustive (in opposed to key members) this should be stated clearly in the text as this explains the large size of the manuscript. Other than this the flow is good and reads clearly.  The use of value judgements to help guide the reader on particularly important information may also be useful.

Length:

Cells has no restrictions on the length of manuscripts, provided that the text is concise. I would argue that while very thorough, the review could benefit from a reduction in size as 21 pages is a little excessive. This would be justified if it was a systematic review but as it is narrative it could do with de-bulking. As a matter of personal preference the addition of extra figures may help break up the large blocks of text.

Figures:

Overall the figures are clear enough, however they are not particularly aesthetically pleasing. As best as one can glean the figures have been produced in Microsoft PowerPoint or similar therefore, the recommendations would be as follows:

General colours

Find a figure with a professional colour palette which use complementary colours selected by graphic designers (nature review is often suitable) and use the ‘Eyedropper’ colour picker tool to sample the fill colours then apply the same palette for all figures.  Pastel colours often work best.

Specific figures

Figure 1.  The diagram is intuitive enough however, the background should be removed from the DNA molecule as it looks copied and pasted in (image is also low res). A potential solve for this would be to identify a suitable free stock image with no background and duplicate the image in the software used to produce the figures. In one of the two duplicated images crop the DNA to the size of the segment which is desired then carefully align the segment over the full-length fragment until flush. The colour of the segment can then be changed under ‘format’ then recolour to highlight it. The image can be grouped and used as usual.

Additionally, it would improve the figure to keep the style consistent, for example for the transcription factors, b) has a shadow (and is therefore 3D) while the DNA and other transcription factors have no shadow effect (2D).

Figure 2. Nothing inherently wrong however, for designing the cell diagrams, if one uses the ‘Curve’ tool instead of the ‘Freeform’ tool to draw structures the outcome is superior when you apply the ‘Bevel’ shape effect, the resulting 3D object will produce a smoother outline without sharp creases (can be somewhat observed in Figure 2). Again, more of a suggestion but creating a semi-transparent 3D cytoplasm using the above method then moving the bevelled nucleus to the back layer of the figure gives the impression of the nucleus contained within the cytoplasm, instead of on top. 

Author Response

Dear Editor,

We sincerely thank you and the reviewers for constructive criticism and suggestions. We appreciated the chance to revise the manuscript. We have taken the remarks into careful consideration. We shortened all the sections and extensively reorganized the manuscript in order to improve the quality of the review. Major text changes have been highlighted in the manuscript with red colour font.

# Response to Reviewer comment No. 1

We thank the Referee for the positive comment.

  1. While the purpose is clearly understood there is no clear rationale for why the discussed transcription factors are being highlighted in opposed to others. It would be good to include this rationale in the final sentence of the introductory paragraph. If this list is exhaustive this should be stated clearly in the text as this explains the large size of the manuscript. The use of value judgements to help guide the reader on particularly important information may also be useful.

In an attempt to exhaustively treat all TFs and relative isoforms described in cancer literature, we fell into a long and narrative review. We decided to reduce the description of those TFs whose splice variants have been characterized only in vitro and therefore lack of strong association data with clinical-pathological variables. We followed the reviewer’s suggestion and at the end of the introduction section we included a final statement that better explains the rationale of reviewing TF isoforms with opposite functions in cancer. Moreover, we divided the text into sections based on the molecular mechanism through which TF isoforms participate to cancer-associated processes. This should better stress important information and guide the reader through the text.

  1. Cells has no restrictions on the length of manuscripts, provided that the text is concise. I would argue that while very thorough, the review could benefit from a reduction in size as 21 pages is a little excessive. This would be justified if it was a systematic review but as it is narrative it could do with de-bulking. As a matter of personal preference the addition of extra figures may help break up the large blocks of text.

In order to reduce the length of the manuscript, we decided to focus on TFs whose splice variants have shown to be informative markers for patients’ stratification in multiple cancer types. The other TFs have been only cited in the background paragraph of each section. All the sections have been reduced. We added extra figures that lighten the review and allow to better point out the molecular mechanisms through which AS TFs can regulate cancer cell phenotypes.

  1. Overall the figures are clear enough, however they are not particularly aesthetically pleasing.

We followed the Reviewer’s suggestions and we hope that the new figures would be more pleasing and appreciated by the Referee.

Reviewer 2 Report

The manuscript „Transcription factors in cancer: when alternative splicing determines opposite cell fates“ by Belluti et al. focuses on alternative splicing events in genes that code for transcription factors and how these events affect tumor development. The topic is timely and very interesting for a broad readership. However, the review is not well done. I am sorry to say this, but it is essentially a long and tiresome list of TF splicing isoforms. Admittedly, I stopped reading after No. 6.

A simple listing of TFs can be helpful when looking up a particular TF (but then it should be in alphabetical order), but not when reading the entire review.  It is also unclear on what basis the order of the TFs was chosen. For example, No. 1 is HELIOS (ikzf2) and No. 3 is IKAROS, after which the protein family was probably named. Why are the two not discussed together?

In my opinion, the readers would benefit from a more detailed description of the molecular mechanisms by which alternative splicing can change TF activity (in analogy to Figure 1), illustrated by 2-3 TF examples for each of the different mechanisms. Furthermore, it would be interesting to focus on 2-3 cancer types and describe how TF isoforms can be used for patient stratification as stated in the abstract. For example as described for NF-Y isoforms in breast cancer.

It should also be added what can be learned from the current knowledge. For example WT1: Lines 207-224 give a list of how the different isoforms of WT1 affect tumor development, with the data being are partly contradictory. What does this tell us? Are the effects cell/tumor-specific? Could it be important to look at the expression levels of all four isoforms as a whole rather than looking at just one? Or could it be important to also consider the expression levels of the interaction partners?

Additional remarks:

  • Please provide a general mechanistic explanation of how cytoplasmic or non-binding TFs act as DN isoforms.
  • Please pay more attention to cite current literature, e.g. line 66: "..., multiple literature data currently show that TFs function as repressors...". This sentence includes no citation, but the next sentence gives two citations on gene repression by TFs from 1989 and 2010. These I do not consider "current". Similarly, line 90: "..., therefore TF-based therapeutic approaches have been proposed as new options for cancer drugs". Publications from the years 1998, 2001 and 2002 are quoted here. Is there no current drug development regarding TFs in cancer?
  • Please make sure to introduce all abbreviations when you use it for the first time, e.g. HDAC, MADS, EMT.

Author Response

We thank the Reviewer for helpful suggestions.

  1. It is essentially a long and tiresome list of TF splicing isoforms. It is also unclear on what basis the order of the TFs was chosen.

We are sorry for the Referee’s comment. We are aware that our effort to comprehensively treat TFs and their cancer-related isoforms led to a very long and too much descriptive story. We reorganized the manuscript and we decided to focus on those TFs whose splice variants have been clearly associated to clinical-pathological variables. General description of TFs have been shortened. TFs have been now ordered based on the molecular mechanism through which AS isoforms exert their antagonistic effect on cancer hallmarks. We divided TFs into three categories: the first one is represented by TFs whose splice variants affect cancer cell phenotype by regulating different classes of genes. The second group is composed by TFs whose splice isoforms activate/repress the same target genes with different efficiency or even opposite effect. The third group encompasses TFs producing DN isoforms that compete with physiological TF activity. In each subclass, TFs have been listed in alphabetical order.

  1. The readers would benefit from a more detailed description of the molecular mechanisms by which alternative splicing can change TF activity, illustrated by 2-3 TF examples for each of the different mechanisms. Furthermore, it would be interesting to focus on 2-3 cancer types and describe how TF isoforms can be used for patient stratification as stated in the abstract. For example as described for NF-Y isoforms in breast cancer.

As described above, we followed the Reviewer’s suggestion and we described molecular mechanisms by which alternative splicing can change TF activity participating to cancer phenotype. For each category, we focused on those TFs associated to clinical parameters.

  1. It should also be added what can be learned from the current knowledge.

In the discussion section, we added a new paragraph that stresses the importance of studying TF splice variants in light of current knowledge of their oncogenic or oncosuppressive antagonistic activity.

  1. Please provide a general mechanistic explanation of how cytoplasmic or non-binding TFs act as DN isoforms.

We added a section dedicated to TF isoforms functioning as DNs. DNs have been further subdivided into three categories on the basis of their mechanistic activity that are:  i) cellular mislocalization, ii) inability to bind target genes or even titration of the functional TF, iii) formation of transcriptional inactive complexes recruited on DNA.   

  1. Please pay more attention to cite current literature. Is there no current drug development regarding TFs in cancer?

We revised cited literature and we added a brief description of the current status of TF-targeting in cancer (see discussion section).

  1. Please make sure to introduce all abbreviations when you use it for the first time, e.g. HDAC, MADS, EMT.

We are sorry for this inaccuracy. All abbreviations have been checked throughout the text.

Round 2

Reviewer 2 Report

Thank you very much for addressing all my concerns so comprehensively. In my opinion, the review is now suitable for publication.